# Survey of Dipeptidyl Peptidase III Inhibitors: From Small Molecules of Microbial or Synthetic Origin to Aprotinin [note 1]

**DOI:** 10.3390/molecules27093006

**Published:** 2022-05-07

**Authors:** Marija Abramić, Dejan Agić

**Affiliations:** 1Ruđer Bošković Institute, 10000 Zagreb, Croatia; 2Faculty of Agrobiotechnical Sciences Osijek, Josip Juraj Strossmayer University of Osijek, 31000 Osijek, Croatia; dejan.agic@fazos.hr

**Keywords:** dipeptidyl peptidase III, DPP III inhibitor, propioxatin, fluostatin, flavonoids, peptidomimetic, benzimidazoles, coumarin derivatives, dipeptidyl hydroxamic acids, guanidiniocarbonyl-pyrrole

## Abstract

Dipeptidyl peptidase III (DPP III) was originally thought to be a housekeeping enzyme that contributes to intracellular peptide catabolism. More specific roles for this cytosolic metallopeptidase, in the renin-angiotensin system and oxidative stress regulation, were confirmed, or recognized, only recently. To prove indicated (patho)physiological functions of DPP III in cancer progression, cataract formation and endogenous pain modulation, or to reveal new ones, selective and potent inhibitors are needed. This review encompasses natural and synthetic compounds with experimentally proven inhibitory activity toward mammalian DPP III. Except for the polypeptide aprotinin, all others are small molecules and include flavonoids, coumarin and benzimidazole derivatives. Presented are current strategies for the discovery or development of DPP III inhibitors, and mechanisms of inhibitory actions. The most potent inhibitors yet reported (propioxatin A and B, Tyr-Phe- and Phe-Phe-NHOH, and JMV-390) are active in low nanomolar range and contain hydroxamic acid moiety. High inhibitory potential possesses oligopeptides from the hemorphin group, valorphin and tynorphin, which are poor substrates of DPP III. The crystal structure of human DPP III-tynorphin complex enabled the design of the transition-state peptidomimetics inhibitors, effective in low micromolar concentrations. A new direction in the field is the development of fluorescent inhibitor for monitoring DPP III activity.

## 1. Introduction

Dipeptidyl peptidase III (DPP III; EC 3.4.14.4; formerly: dipeptidyl arylamidase III, dipeptidyl aminopeptidase III) is a proteolytic enzyme that catalyzes the hydrolytic cleavage of dipeptides sequentially from the N-termini of its peptide substrates. It was discovered in extracts of bovine pituitary gland as the third in a series of dipeptidyl arylamidases through the hydrolysis of synthetic substrate Arg-Arg-2-naphthylamide [1]. The enzyme was further purified and biochemically characterized from the cytosolic fraction of several human and animal tissues and lower eukaryotes [2]. Studies in vitro have demonstrated that DPP III prefers the Arg-Arg-arylamide among synthetic substrates, but displays a relatively broad specificity towards peptide substrates, optimally sized from tetra- to octa-peptides [1,3]. A high affinity of DPP III was shown for biologically active peptides, anigotensins, enkephalins and endomorphins [3,4,5]. Investigation with general protease inhibitors revealed that DPP III is a metallopeptidase sensitive to thiol-blocking agents [2]. In addition, earlier biochemical and histochemical research of DPP III activity and protein content showed its broad tissue distribution, and enhanced level in endometrial and ovarian malignancies [6,7].

Structure–activity relationship study of DPP III began when the genes encoding rat and human enzymes were cloned and their amino acid sequences deduced [2,8]. The experimental evidence on the presence of one Zn^2+^ ion per molecule of human placental and rat recombinant enzyme, together with low dissociation constant (250 fM), proved that DPP III is a zinc-metallopeptidase [8]. The flow of data on primary protein structures, due to the whole genome sequencing, similarity search and multiple sequence alignment, led to the discovery of bacterial DPPs III and recognition of the unique DPP III family, also named metallopeptidase family M49 in the MEROPS database [9]. The M49 family is defined with five evolutionarily conserved regions, including motifs HEXXGH and EEXR(K)AE(D) that contain the residues important for zinc binding and catalytic activity [8,9,10,11]. The next breakthrough in the research of DPP III was a resolution of the crystal structure of yeast and human DPP III [12,13] which enabled the application of computational methods and study of protein flexibility and catalytic mechanism [14,15,16]. The crystal structure of DPP III features an elongated protein molecule with two lobe-like domains separated by a wide cleft (Figure 1). The zinc-binding site and the catalytically important residues are located in the upper lobe, which is mostly α-helical. The lower lobe, besides having α-helices, contains a smaller β-sheet portion (a five-stranded β-barrel). The two domains are connected by a number of loop regions. The first 3-D structure of human DPP III in complex with pentapeptide revealed a huge domain motion resulting in the complete closure of the cleft around the bound peptide substrate [13] (Figure 1). The active-site zinc ion is coordinated by the two histidine residues that belong to the conserved HEXXGH motif, a second glutamic acid residue from the conserved sequence motif EEXR(K)AE(D) and by the water molecule. In the structure of human DPP III, zinc-ligands are His450, His455 and Glu508. The Glu from the first zinc-binding motif (Glu451 in human DPP III) is proposed to act as a general base activating the water molecule which attacks the scissile peptide bond [12,16].

Direct evidence about the biological functions of DPP III was elusive for a long time, partly due to the lack of specific inhibitors. Metal chelating agents and sulfhydryl blocking reagents are known non-specific inhibitors of DPP III from various tissue sources [2]. Inhibition with EDTA and 1,10-phenanthroline is consistent with the metallopeptidase nature of DPP III. Among the sulfhydryl reagents, the most potent inhibitors are organomercurial compounds PCMS, PCMB and pHMB, effective in micromolar concentration. Strong inhibition caused by some thiol reagents and reversal arising from the addition of thiol compounds suggested that SH-group(s) are essential for DPP III catalytic activity. However, mutational analyses and 3-D structure elucidation did not support this hypothesis [12,13,17]. Comparative biochemical studies have shown that DPP III sensitivity to thiol blockers varies with species: e.g., all tested reagents inhibited the rat erythrocyte enzyme with higher potency than the human erythrocyte enzyme [18]. Rat DPP III was hyperreactive to organomercurial pHMB whose IC_50_ was 3 nM, and the same reagent produced 50% inhibition of human DPP III activity at about 1µM concentration [18,19]. Reactive cysteine residues, responsible for DPP III inactivation by sulfhydryl reagents, were identified by site-directed mutagenesis studies conducted for the rat (Cys176), yeast (Cys518, Cys639), human (Cys19, Cys176, Cys654) and bacterial enzyme (Cys450) [17,20,21,22]. Interestingly, while reactive cysteine is a part of the active–site motif HEC450LGH of the bacterial enzyme, Cys176, responsible for inactivation of human and rat DPP III by the organomercurial compound, is a residue 44 Å apart from the catalytic center of the ligand-free human DPP III. Modification of Cys176 probably hinders the formation of the closed active site upon substrate binding.

Participation in the final steps of intracellular protein catabolism was suggested for DPP III based on its cytosolic localization, wide distribution and broad specificity for oligopeptides. The revival of research interest in this enzyme in the last decade confirmed the suspected role of DPP III in post-proteasomal cleavage of peptides [23] and revealed several more specific roles. Most important is DPP III’s role in the degradation of biologically active peptides of the renin–angiotensin system—angiotensin II to IV, and angiotensin (1–7)—and endogenous defense against oxidative stress [24,25,26]. Participation of DPP III in oxidative stress regulation in mammalian cells does not depend on its enzymatic activity but is related to its ability to bind to the ubiquitin ligase Keap1, a constituent of the Keap1–Nrf2 signaling pathway, the cell’s main defense mechanism against environmental toxins and carcinogens. The molecular mechanism of interaction of DPP III-Keap1 has been recently reported [27].

DPP III expression is dysregulated in several cancers [7,26,28]. In ovarian primary carcinomas, DPP III activity correlates with tumor aggressiveness [29]. Overexpression of DPP III was shown to be correlated with poor prognosis of human breast cancer patients [30] and colorectal cancer patients [28]. According to recent findings, the pathophysiological role of DPP III in malignant growth seems to be connected to its interaction with Keap1, leading to upregulation of the Keap1-Nrf2 pathway [26,30,31] and with cyclin-dependent kinase 1 (CDK1) [28]. Most recent studies using the luminometric immunoassay method have shown that DPP III is elevated in the plasma of septic patients, and that the highest levels of circulating DPP III are present in non-survivor septic shock patients [32]. Therapeutic potential of circulating DPP III inhibition by the specific antibody Procizumab in the restoration of altered cardiac function during sepsis is under investigation in preclinical sepsis models [32,33].

DPP III in vitro hydrolyzes pentapeptides Leu- and Met-enkephalin at the Gly2-Gly3 bond as so-called enkephalinase B, a peptidase at first isolated from the rat brain membranes by detergent treatment [34]. However, the identity of enkephalinase B and DPP III is not yet proven by biochemical and/or structural data. Also, there is no conclusive evidence on the role of DPP III in the endogenous pain-modulatory system, although this enzyme in high concentrations colocalizes in the neurons of the rat spinal dorsal horn with endogenous opioid peptides enkephalins and endomorphins [35]. Earlier findings indicated an association of DPP III with cataract formation [36] and, more recently, with influenza A virus infection [37]. Obviously, elucidation of the (patho)physiological relevance of DPP III, especially of the human enzyme, requires further investigations, for which specific and effective inhibitors would be necessary. In this review we present an historical perspective and the most current strategies for the discovery or development of DPP III inhibitors, with special emphasis on small molecules and their mechanism of inhibitory action. Inhibitor potency is expressed as the IC_50_ value, defined as the concentration of an inhibitor which causes 50% reduction of enzyme activity, or by the inhibition constant K_i_, the equilibrium dissociation constant of the enzyme-inhibitor complex.

## 2. Inhibitors of Microbial Origin

### 2.1. Peptide Aldehydes

Microorganisms produce low-molecular-mass secondary metabolites exhibiting large structural diversity and various biological activities. An early initiative directed towards the discovery of DPP III inhibitors from microbial cultures resulted in the isolation of acetyl-L-leucyl-L-argininal. This compound, isolated from the culture filtrate of a bacterium BMG 520-YF2, inhibited DPP III purified from the rat pancreas with an IC_50_ of about 0.1 µM [38]. It is structurally related to another microbial peptide aldehyde, leupeptin, Ac-Leu-Leu-argininal. Known as a non-specific inhibitor of cysteine (and serine) peptidases, leupeptin forms a thiohemiacetal intermediate with a catalytic cysteine residue. Potent inhibition (IC_50_: 0.06 µM) of rat pancreas DPP III by leupeptin is reported [38], but its inhibitory potency for human DPP III is much lower (100 µM leupeptin caused 59% inhibition of enzyme activity) [3].

### 2.2. Fluostatins

The same approach of screening revealed fluostatins A and B, compounds containing a tetracyclic nucleus with fluorenone skeleton (Figure 2), isolated from the fermentation broth of a *Streptomyces* sp. TA-3391, as new inhibitors of human DPP III purified from placenta [39]. The IC_50_ values of fluostatins A and B were 1.4 µM and 74 µM, respectively, with the synthetic substrate Arg-Arg-2NA. Fluostatin A inhibited the hydrolysis of peptide substrate Leu-enkephalin by DPP III with the K_i_ value of 14.2 µM and the inhibition was mixed type. Both fluostatins were only slightly inhibitory against DPP I, DPP II, and DPP IV [39,40].

### 2.3. Propioxatins

Highly potent inhibition of DPP III purified from the soluble fraction of the rat brain was shown with propioxatin A (K_i_: 13 nM) and propioxatin B (K_i_: 5.6 nM), two compounds isolated from the culture filtrate of the actinomycete *Kitasatosporia setae* [41,42]. Both propioxatins were discovered as inhibitors of enkephalinase B from rat brain membranes [42] and structurally elucidated to consist of a dipeptide *N*-acyl-L-Pro-L-Val, with *N*-acyl containing hydroxamic acid [43]. *N*-acyl moieties were determined to be α-propyl and α-isobutyl succinic acid β-hydroxamic acid for propioxatin A and B, respectively (Figure 3). Inhibitory activity of propioxatins was assayed further for a number of peptidases, including enkephalinase A (endopeptidase which cleaves Gly^3^-Phe^4^ bond in enkephalins), trypsin, chymotrypsin, thermolysin, papain, carboxypeptidase A and B. Pronounced inhibition was observed only with enkephalinase B (IC_50_ with propioxatin A and B: 0.036 µM and 0.34 µM, respectively), leucyl-aminopeptidase (IC_50_ with propioxatin A: 2.6 µM) and aminopeptidase M (IC_50_ with propioxatin A: 0.39 µM) [42].

### 2.4. Other Protease Inhibitors of Microbial Origin

Microbial compounds which were isolated during the course of extensive screening programs for inhibitors of other peptidases (aminopeptidases or endopeptidases), such as amastatin, bestatin, puromycin, pepstatin and chymostatin, do not affect DPP III activity significantly when added in concentrations that strongly inhibit their target enzymes (e.g., 100 µM) [19].

## 3. Inhibitors of Plant Origin

### 3.1. Flavonoids

Flavonoids are a diverse group of polyphenolic phytochemicals ubiquitously found in fruits and vegetables. They exhibit a wide range of biological activities playing an important role in growth, development and defense of plants. Flavonoids are also constituents of human diet with potential health benefits, due to the antioxidative effects (scavenging of free radicals) attributed to their phenolic hydroxyl groups. As they are effectors on various biological systems, flavonoids are capable of modulating the activity of enzymes [44]. The inhibitory effect of flavonoids towards human DPP III was investigated by a combination of experimental and computational methods. Out of 15 flavonoids tested (Table 1), the most effective were luteolin (flavone group), galangin and fisetin (both from the flavonol group) which inhibited DPP III hydrolytic activity towards Arg-Arg-2NA with an IC_50_ ~20 µM [45]. The number and exact distribution of hydroxyl groups on the flavonoid core (consisting of two phenyl rings, A and B, joined with pyran ring C) were important for the inhibitory properties of tested flavonoids.

According to 3D QSAR modeling, the presence of hydrophilic regions at a flavonoid molecule increases its inhibitory activity. Molecular dynamic (MD) simulations, performed for luteolin, confirmed the importance of hydroxyl groups for binding onto the active site of human DPP III. Interestingly, flavonoid compounds do not bind to the active-site zinc ion, but rather to the hydroxyl groups on its A and B ring, mostly forming multiple interactions (H-bonds, van der Waals, electrostatic) with amino acid constituents of S2, S1 and S1′ subsite of human DPP III [13]. In addition, interactions of B-ring (CH-π) and carbonyl group of luteolin C-ring (H-bond) with His568, a crucial constituent of the DPP III active site, are revealed during MD simulations (Figure 4) [45].

### 3.2. Polyphenolic Compounds in Plant Extracts

Agić et al. [45] reported that two flavonoid compounds from the flavonol group, kaempferol and quercetin, inhibited in vitro hydrolytic activity of human DPP III with IC_50_ values of 32.9 µM and 74.1 µM, respectively (Table 1). Experimental research on plant extracts has revealed that dominant flavonols in fruits of the genus *Prunus* include kaempferol and quercetin in the form of glycosides [46], and quercetin 3-O-rutinoside was found as the most dominant. Popović et al. [46], investigating the extracts prepared from the fruits of different *Prunus* species, have found the highest bioactive potential in blackthorn and steppe cherry. Except for the highest antioxidant capacity, extracts from the fruits of these two *Prunus* species showed also the highest antiproliferative effect in human tumor cells HT29 and in vitro inhibition activity towards α-amylase, α-glucosidase and human DPP III. Inhibitory activity towards the human DPP III was positively correlated with antioxidative potency, antiproliferative effect and with some polyphenols/flavonoids of the blackthorn and steppe cherry extract. Further in silico analysis (calculation of binding energies and molecular docking) pointed out quercetin 3-O-rutinoside as the most effective inhibitor of DPP III [46]. Docking and prediction with LigProt+ indicated binding of quercetin 3-O-rutinoside in a pocket near the enzyme active site through the H-bonds and hydrophobic interactions which engage several constituents of the S1′, S1 and S2 subsite, including three for activity essential residues (His 568, Glu451 and His450) (Figure 5).

The same approach for determining and correlating phytochemicals and biological activities was applied in investigation of the extracts of the cornelian cherry fruit, a rich source of iridoids and polyphenols [47]. A number of biological activities were found to be expressed, including high antioxidant capacity and high inhibitory potential for human DPP III. By principal component analysis (PCA), the constituents with the highest contribution to experimentally determined bioactivities were shown. Analysis with in silico methods distinguished polyphenolic compound pelargonidin 3-robinobioside as the best inhibitor of human DPP III in the fruit extract of cornelian cherry [47]. Docking predicted that this anthocyanidin compound interacts with three residues from the DPP III active site (His568, Glu451 and His450) through hydrogen bond formation and displays many hydrophobic contacts with amino acid residues of the DPP III substrate binding site (Figure 5).

## 4. (Poly)Peptide Inhibitors

### 4.1. Dipeptides

Dipeptides are products of hydrolytic reactions catalyzed by DPP III. Inhibition of DPP III with a dipeptide, which was first observed for the bovine pituitary enzyme and Arg-Arg [1], seems to be dependent on the origin of DPP III. Thus, rat brain DPP III was potently inhibited with dipeptides with aromatic pairs, Tyr-Tyr and Tyr-Phe with the inhibition constants of 5.8 µM and 8.4 µM, respectively, while the K_i_ value for Arg-Arg was 35.7 µM [4]. The activity of human erythrocyte DPP III was only slightly decreased by dipeptides containing at least one aromatic amino acid residue [3]. Leu-Trp exerted the most potent inhibitory effect (K_i_ value 39 µM) on the goat brain DPP III among 27 tested dipeptides [48]. In contrast, Leu-Trp inhibited human DPP III with a K_i_ value of 128 µM [3].

### 4.2. Oligopeptides

Natural heptapeptide spinorphin, Leu-Val-Val-Tyr-Pro-Trp-Thr, isolated from the bovine spinal cord as a potent endogenous inhibitor of enkephalin-degrading enzymes, and its truncated form, the synthetic pentapeptide tynorphin (Val-Val-Tyr-Pro-Trp) were reported to be competitive inhibitors of DPP III from a monkey brain cytosolic fraction, with K_i_ values of 2.42 µM and 0.075 µM, respectively [49]. Spinorphin, in contrast to tynorphin, which was a more specific inhibitor of DPP III, exhibited potent inhibitory activity towards several other enkephalin-degrading enzymes, such as aminopeptidase, neutral endopeptidase and angiotensin-converting enzyme. However, it was shown that both oligopeptides are easily degraded in the body, primarily by the hydrolytic action of an aminopeptidase [49,50]. Tynorphin and a heptapeptide valorphin (Val-Val-Tyr-Pro-Trp-Thr-Gln) isolated from bovine hypothalamic tissue, potently inhibited hydrolysis of Arg-Arg-2NA catalyzed by human DPP III (K_i_ values of 4.3 nM and 49 nM, respectively) [5,51]. Valorphin was shown to be cleaved by human DPP III very slowly. In addition, Leu-valorphin-Arg is a reported peptide substrate of this enzyme [5]. Therefore, “oligopeptide inhibitors” spinorphin, valorphin and tynorphin, members of hemorphin group of peptides should be considered also as competing substrates of DPP III. The amino acid sequences of hemorphins show homology with residues 32–38 of the β-chain of bovine hemoglobin. To find the most effective inhibitor of rat recombinant DPP III, a series of hemorphin-like pentapeptides with aliphatic or aromatic amino acids at the N-termini were synthesized by Chiba et al. [35]. The strongest inhibitory activity showed Ile-Val-Tyr-Pro-Trp and Trp-Val-Tyr-Pro-Trp, with K_i_ values of 0.1 µM and 0.126 µM, respectively, while tynorphin exhibited somewhat weaker inhibition (K_i_: 2.67 µM) of the rat DPP III. 

The crystal structure of human DPP III (inactive mutant E451A) in complex with tynorphin has been solved revealing the formation of the closed enzyme’s active site due to large domain movement during peptide binding [13] (Figure 1). This structure also exposed the architecture of human DPP III substrate binding site as being composed of five deep subsites (S2 to S3′) consisting of seven (S2′) to thirteen (S2) amino acid residues. Domain motion brings the catalytic zinc ion close to the carbonyl group of the scissile (penultimate) peptide bond between P1 and P1′ according to Schechter and Berger nomenclature. Microcalorimetric analyses have shown that the binding of tynorphin is an entropy-driven process [13]. The release of ordered water molecules from the binding cleft is proposed to represent the entropic driving force of the peptide substrate binding to the enzyme. The bound peptide is completely buried between the two domains of the DPP III protein. The N-terminus is anchored by polar interactions to the side chains of Glu316 and Asn394 and to the carbonyl group of Asn391. The C-terminal carboxylate of tynorphin forms a salt bridge with Arg669. Polar interactions in the complex are formed almost exclusively with residues of the lower lobe of DPP III, with the exception of the H-bond between His568 and the carbonyl group of P1 residue (Figure 6).

High-resolution crystal structures of complexes of human DPP III inactive variant (E451A) with good in vitro substrates (pentapeptides Leu- and Met-enkephalin, tetrapeptide endomorphin-2, octapeptide angiotensin II) and the peptide “inhibitor” Ile-Val-Tyr-Pro-Trp confirmed the large conformational change of the DPP III protein upon ligand binding, and revealed conservation of peptide-binding mode in the inter-domain cleft of this enzyme [52]. Common to all complex structures were extensive polar interactions of the peptide’s N-terminal residues and the backbone interactions of the peptide via hydrogen bonds with the core of DPP III, which ensure the correct positioning of the scissile peptide bond to the catalytic zinc ion. Interestingly, a difference in the coordination of the carbonyl group of the scissile bond is observed. In the complexes with Leu- and Met-enkephalin, this carbonyl group does not interact directly with the zinc ion, but a water molecule completes the tetrahedral coordination of the metal ion. On the other hand, the active-site zinc ion is coordinated by the respective carbonyl oxygen in the complexes with endomorphin-2 and “inhibitory” peptide Ile-Val-Tyr-Pro-Trp.

The binding mode of peptides observed in the available crystal structures does not allow a clear distinction between the peptides which are good substrates of DPP III and those which are reported as substrate “inhibitors” or “slow” substrates. However, most recently, by using quantum molecular mechanics calculations and various molecular dynamics techniques, Tomić and Tomić [53] described the entire catalytic cycle of human DPP III and explained why tynorphin is a poor substrate compared to Leu-enkephalin. They did not find a difference in the reaction mechanism, but in significantly higher stabilization of the intermediate structure and of the products of tynorphin hydrolysis in the active site, which hinder regeneration of the enzyme [53].

### 4.3. Peptidomimetic Inhibitors

Only recently, by using the knowledge of 3-D structure of complex tynorphin-inactive human DPP III, Ivković et al. [54] designed peptidomimetic inhibitors. They synthesized (*S*)- and (*R*)-epimers of hydroxyethylene transition state mimetics of tynorphin, named SHE and HER (Figure 7), and characterized their potentials as inhibitors of human DPP III. The non-cleavable hydroxyethylene isostere was used to replace the scissile penultimate peptide bond in pseudo-pentapeptides mimicking the tynorphin. Both SHE and HER inhibited the enzyme with an IC_50_ of 98.5 µM and 13.8 µM, respectively. As for tynorphin, an endothermic binding to human DPP III was observed with K_d_ values of 23 µM for SHE and 11 µM for the binding of HER [54]. The solved crystal structure of the complex SHE-human DPP III (E451A) revealed almost the same binding mode of this pseudo-pentapeptide inhibitor as that of tynorphin [13]. The equivalence to the tynorphin–human DPP III complex was observed at the N-terminus of SHE, which was also tightly bound to the enzyme by three hydrogen bonds (to Glu316, Asn391 and Asn394) and a salt bridge to the Glu316. Similar to tynorphin, the C-terminal part of the ligand SHE was bound in a cation-π complex of its indole moiety with Lys670 and Arg669, and a salt bridge was formed between the Arg669 and the C-terminal carboxylate group of bound SHE. Unlike tynorphin, (*S*)-epimer (SHE) did not interact with His568 (in the complex with tynorphin this amino acid residue forms a hydrogen bond with the carbonyl group of Val-2). However, its hydroxyl substituent complexes to the zinc ion (the length of Zn-O coordinating bond was found 1.9 Å) as expected. Ivković et al. [54] have predicted that (*R*)-hydroxyehylene would form both a coordinative bond with the catalytic zinc ion and a hydrogen bond to His568, thus explaining the stronger inhibitory effect of (*R*)-epimer (HER).

### 4.4. Aprotinin

Human DPP III hydrolytic activity is competitively inhibited by aprotinin (a 58 amino acid long, strongly basic, polypeptide), a natural inhibitor of serine endopeptidases, also known as bovine pancreatic trypsin inhibitor (BPTI), with a K_i_ value of 11.7 µM at physiological pH [51]. Interaction of aprotinin with human DPP III has been studied by experimental and molecular modeling approach [55]. Aprotinin was examined as a potential substrate of DPP III, since four amino acid residues at its N-terminus (Arg-Pro-Asp-Phe) are extended from the compact globular structure of this polypeptide, and human DPP III possesses also post-proline cleaving ability [5]. However, no degradation was detected by MALDI-TOF MS analysis, even after long incubation of aprotinin with a high concentration of human DPP III [55], in agreement with previous investigations showing that polypeptides are not substrates of DPP III. A molecular modeling approach (docking and long molecular dynamics simulations), based on the knowledge of the crystal structures of human DPP III in its open form and bovine aprotinin, revealed that this polypeptide interacts with the substrate binding cleft of the enzyme by its canonical binding loop, where the free N-terminus (Arg-Pro-Asp-Phe) is positioned distant from the enzyme (Figure 8) which explains the absence of enzymatic cleavage [55]. In contrast to DPP III, aprotinin is in vitro substrate and has been shown to interact with the active center of dipeptidyl peptidase IV (DPP IV) by its free N-terminus (Arg^1^-Pro^2^ residues) [56].

Analysis of DPP III–aprotinin interactions during MD simulations has revealed that amino acid residues lining the human DPP III interdomain cleft make numerous H-bonds and hydrophobic interactions with the inhibitor (Figure 8), including aprotinin’s canonical binding loop (residues ^13^PCKAR^17^). As seen in Table 2, most of the H-bonds were formed between Lys15 and Arg17 from the aprotinin binding epitope (important for inhibitory function with serine endopeptidases) with amino acid residues that are constituents of human DPP III S1, S1′, S2, S2′ and S3′ substrate binding subsites [13,52], and with Ser101 and Ser384. In addition to hydrogen-bonding and hydrophobic interactions, for binding of aprotinin to human DPP III (i.e., strong stabilization of DPP III–aprotinin complex) intermolecular electrostatic interactions Arg39-Asp372, Lys41-Asp496 and Lys46-Asp396 (written as aprotinin–human DPP III amino acid pair, respectively) [55] are very important. The importance of aprotinin interaction with Asp496 for inhibitory binding to human DPP III was shown experimentally by mutational analysis when the replacement of this amino acid residue with Gly lowered inhibitory potency of aprotinin by 12.7-fold (increase of a K_i_ value for D496G variant compared to the K_i_ value for the wild-type human DPP III) [51]. Calculations of electrostatic potential indicated that the positively charged aprotinin molecule is situated just above the negatively charged DPP III region next to the five-stranded β-core of the DPP III lower domain, while negatively charged aprotinin residues close to its C- and N-terminus are sticking outside of the protein interdomain cleft [55].

## 5. Synthetic Inhibitors

### 5.1. Benzimidazole Derivatives

A small set of benzimidazole derivatives was synthesized and screened as potential DPP III inhibitors [57]. The benzimidazole moiety was chosen due to its reported potential of H-bonding and π-π stacking interactions with the imidazole ring of His residues essential for the activity of some metallo- and serine peptidases. Basic amidino groups were added to mimic the free amino group required in DPP III substrates. Indeed, the presence of an amidino group was essential for obtaining potent DPP III inhibition by benzimidazole derivatives, and the cyclization of amidino functionalities significantly increased the inhibitory potency (Table 3). To further improve the inhibitory potential, a cyclobutene ring was introduced as rigid conformation support to the diamidino substituted dibenzimidazoles. Two cyclobutene derivatives containing amidino-substituted benzimidazole moieties were found to inhibit DPP III strongly, with IC_50_ value below 5 µM (Table 4) [57].

The inhibition with one representative, compound **1′**, 1,3-di-[5-(2-imidazolinyl)-2-benzimidazolyl]-2,4-di-phenyl-cyclobutane dihydrochloride, was studied in more detail. Compound **1′** inactivated human DPP III time–dependently and the peptide substrate (valorphin) exerted protective role, while Co^2+^ and Zn^2+^ ions were ineffective, showing that the inhibitor does not act via non-specific chelation of the active-site metal ion [57]. The obtained second-order rate constant for inactivation of human DPP III by compound **1′** (6924 M^−1^min^−1^) compares well with the second-order rate constant for inactivation of the same enzyme (human erythrocyte DPP III) with sulfhydryl reagent *p*-hydroxy-mercuribenzoate (3523 M^−1^min^−1^) [18]. The inhibitory mechanism with compound **1′** was further studied by MD simulations which confirmed that the binding of this inhibitor has no effect on the coordination of the active-site zinc ion [58]. Analysis of enzyme-inhibitor interactions revealed that compound **1′** interacts with DPP III mostly electrostatically, the strongest interactions being those between imidazolinyl groups of inhibitor and the amino acid residues of the lower (Ser108, Gly110, Tyr318 and Ala416) and upper (Tyr417, Asn545 and Glu667) domains (Figure 9). Benzimidazole groups interacted with Tyr318 and Tyr417, and the phenyl groups and cyclobutane ring interacted with the enzyme through a combination of van der Waals and electrostatic interactions. The cyclobutane ring interacted with His568. As seen from Figure 9, the binding of compound **1′** engaged Tyr318 and His568, two evolutionary conserved amino acid residues in the DPP III family, which are crucially important for DPP III hydrolytic activity. The binding mode and the multiple interactions established during the MD simulations of the potent benzimidazole-based inhibitor in complex with human DPP III, differ from those seen for the reversible inhibitor (substrate analogue Tyr-Phe-NHOH) [14] and pentapeptide tynorphin [13], which could explain the experimentally observed irreversible (tight binding) inhibition of DPP III by this compound.

### 5.2. Analgesic and Antihypertensive Drugs

DPP III purified from goat brain is considered to be a functional homologue of mammalian DPP III since it possesses a lower molecular weight (69,000) [48]. The inhibitory potency of several analgesic and antihypertensive drugs was tested for goat brain DPP III in an in vitro assay with Arg-Arg-4-methoxy-β-naphthylamide as substrate [59]. Competitive inhibition was observed with diclofenac (K_i_: 0.3 mM), paracetamol (K_i_: 0.25 mM) and rofecoxib (K_i_: 0.19 mM), while analgesic nimesulide exhibited a mixed type of inhibition (K_i_: 70 µM). Antihypertensive drugs atenolol, metroprolol and labetalol inhibited the enzyme in a competitive manner with K_i_ value of ~0.2 mM [59].

### 5.3. Coumarin Derivatives

Coumarin and its derivatives are a group of oxygen-containing heterocyclic compounds with benzopyrone skeleton. These plant-derived, or synthetically obtained, substances possess a wide variety of biological activities (antioxidant, anti-inflammatory, antiviral, anticancer). Most recently, 40 synthetic coumarin derivatives were examined as potential inhibitors of human DPP III in the study which combined in vitro hydrolytic activity determination with the molecular modeling (QSAR, docking and MD simulations) methods [60]. Out of 40 tested compounds, 13 coumarin derivatives at a concentration of 10 µM inhibited the DPP III hydrolytic activity by 20% or higher (Table 5). The most potent inhibitor was compound **12**, i.e., 3-benzoyl-7-hydroxy-2H-chromen-2-one (IC_50_: 1.1 µM).

Docking predicted that compound **12** binds to the inter-domain cleft of human DPP III. Importance of the H-bonds between the 7-OH group of this coumarin derivative and the carbonyl group of Glu329, and the van der Waals interactions with Ile315, Ser317, Phe381, Pro387 and Ile390 for binding were revealed by the MD simulations. Based on the results of the QSAR analysis, structures of two new coumarin compounds with improved inhibitory activity were proposed [60] (Figure 10).

### 5.4. Guanidiniocarbonyl-Pyrrole-Aryl Conjugates

The structure of synthetic DPP III substrate, Arg-Arg-2-naphthylamide, having a guanidine and fluorescent aryl unit, was the starting point in the design of potential DPP III inhibitors consisting of a guanidiniocarbonyl-pyrrole (GCP) unit and fluorophore (pyrene). The GCP moiety was chosen as a similar structure, in which aliphatic guanidine was replaced with more acidic pyrrole guanidine and the aryl group was larger (pyrene). The development of a fluorescent inhibitor is interesting because of the possibility of monitoring DPP III action in vitro and in vivo.

First examination of pyrene-GCP conjugate showed promising DPP III inhibition properties [61]. In new series of six pyrene-GCP conjugates (designated **A** to **F**), linker length and rigidity were systematically varied as well as the charge and steric properties by the introduction of lysine [62]. By fluorometric titration experiments, binding constants for complexes with human DPP III (inactive mutant E451A) and fluorescence change were determined. The fluorometric response of various pyrene derivatives upon binding to DPP III strongly varied, from emission increase to emission quenching. Interestingly, the binding affinity of all pyrene-GCP conjugates for DPP III was within the same order of magnitude. However, only two such compounds (**D** and **F**) (Figure 11) potently inhibited DPP III activity, both with K_i_ value ~0.3 µM. Interactions between the human DPP III and pyrene-GCP conjugates were studied by MD simulations. The computational study suggested that non-inhibitory compounds bind differently to DPP III, not interfering with the Arg-Arg-2NA hydrolysis, while inhibitory compound **F** establishes strong electrostatic interaction with Glu451 and His568 of the DPP III active site [62].

As an alternative to pyrene, cyanine dyes were used as polarity-sensitive fluorimetric probes in GCP conjugates, as they are intrinsically nonfluorescent but show strong fluorescence when bound to the target. A preliminary study showed that cyanine-GCP conjugate (Cy-GCP) is an efficient inhibitor of human DPP III [63]. An analogue of Cy-GCP, with reversed connectivity of cyanine dye (over benzothiazolium part) and with extended linker, was prepared [62] (Figure 11). This intrinsically nonfluorescent cyanine-GCP conjugate (CIA) showed strong emission upon binding to human DPP III (E451A) and micromolar binding affinity, 30 times stronger compared to previously examined close analogues. Enzyme kinetic studies revealed CIA as a potent competitive inhibitor of human DPP III (K_i_ value at pH 7.4: 0.228 µM), and MD simulations pointed to its strong interaction with two constituents of the DPP III active site, Glu451 and His 568. The results of Ćehić et al. [62] point towards cyanine-GCP analogues as promising lead compounds for simultaneous monitoring and inhibiting of DPP III.

Ban et al. [64] prepared triarylborane dyes that bind to proteins (BSA and human DPP III) with high affinity, exhibiting up to 100-fold increase in fluorescence. Interestingly, these compounds did not impair enzymatic activity of DPP III; thus, being the fluorimetric markers for this protein.

### 5.5. Dipeptidyl Hydroxamic Acids

Hydroxamic acid derivatives are among the most potent inhibitors of metallopeptidases due to their ability to form bidentate complexes with the active-site metal ion. Dipeptidyl hydroxamic acids with free N-terminal amino group were chosen as substrate analogues for binding to S2 and S1 subsites of human DPP III and were shown to be potent competitive inhibitors of this enzyme; useful for the study of its active site [65,66]. Interestingly, a dramatic difference in the inhibitory potency was observed, depending on the P1 residue of the dipeptidyl hydroxamic acid inhibitor. Thus, H-Tyr-Phe-NHOH inhibited human DPP III with the K_i_ value of 0.15 µM, but H-Tyr-Gly-NHOH was ~70-fold less potent inhibitor (K_i_ of 10.5 µM at pH 8.0), indicating that for strong interactions with human DPP III the chelating effect of hydroxamate moiety is not sufficient. Yet, interactions between the amino acid residues at P1 and P2 positions of the inhibitor and the corresponding S1 and S2 enzyme subsites are crucial for inhibitory potency [66]. A small series of dipeptidyl hydroxamic acids were synthesized and investigated in this respect: H-Phe-Phe-NHOH, H-Phe-Leu-NHOH and H-Phe-Gly-NHOH [67]. The pH value of the enzyme reaction medium strongly influenced the inhibitory activity of all examined dipeptidyl hydroxamates: the K_i_ values were 2-3 times lower at pH 7.4 compared to values determined at pH 8.0. As seen in Table 6, the most potent inhibitor of this series was the compound with Phe as side chain substituent in P1 position, which inhibited human DPP III with equal potency (K_i_ ~0.1 µM at pH 8.0 and K_i_ ~0.03 µM at pH 7.4) as H-Tyr-Phe-NHOH, while H-Phe-Leu-NHOH (K_i_ at pH 8.0: 1.24 µM; K_i_ at pH 7.4: 0.65 µM), and especially H-Phe-Gly-NHOH (K_i_ at pH 8.0: 14.5 µM; K_i_ at pH 7.4: 4.6 µM) was much weaker inhibitor [67].

MD simulation of the human DPP III-H-Tyr-Phe-NHOH complex revealed a monodentate mode of binding for this dipeptidyl hydroxamate inhibitor to the active-site zinc ion (Figure 12) [14]. Namely, the second carbonyl group from the inhibitor N-terminus interacts to zinc, while its hydroxamate hydroxyl group forms a hydrogen bond with Glu508 from the active site. In addition, another active-site glutamate, Glu451, is hydrogen-bonded to the amide group of the first peptide bond of the inhibitor and to its N-terminus (Figure 12).

### 5.6. Hydroxamate Inhibitor JMV-390

A non-specific inhibitor of several metallopeptidases *N*-[3-[(hydroxyamino)carbonyl]-1-oxo-2(R)-benzylpropyl]-L-leucine (JMV-390) was shown to block, highly potently, hydrolytic activity of human recombinant DPP III with angiotensin (1–7) as substrate (IC_50_ of 1.4 nM), and helped identification of DPP III as the angiotensin (1–7) degrading peptidase in human renal epithelial cells HK-2 [25] (Figure 13).

## 6. Conclusions

This review encompassed a whole range of natural and synthetic (low molecular mass) compounds with experimentally proven in vitro inhibitory activity toward mammalian DPP III. Whenever known, the molecular basis for observed inhibition was explained.

Although DPP III belongs to the metallopeptidase class of proteolytic enzymes, it is inhibited with a low micromolar concentration of some sulfhydryl reagents due to the reactivity of its cysteine residues and high flexibility of its 3-D structure.

Early searches for specific inhibitors of DPP III activity, based on the screening of microbial culture filtrates, yielded several low molecular mass secondary metabolites, with propioxatin A and B as the most potent compounds (K_i_: 13.0 nM and 5.6 nM). Up to now, propioxatin B is one of the strongest inhibitors of DPP III yet reported. DPP III inhibitors were further recognized among polyphenolic compounds (flavonoids and their glycosides), and coumarin and benzimidazole derivatives. Natural polypeptide aprotinin is also on the list of DPP III inhibitors in vitro (K_i_: 11.7 µM).

High inhibitory potential for DPP III was revealed in DPP III substrate analogues, dipeptidyl hydroxamic acids (K_i_ values in nM range), and in oligopeptides from the hemorphin group (valorphin, tynorphin). However, the latter are actually “slow” substrates of DPP III in vitro, susceptible to rapid degradation in vivo by aminopeptidases. Recently, the crystal structures of human DPP III, ligand-free and in complex, have become accessible allowing, combined with computational methods, the elucidation of inhibition mechanisms and design of improved, more specific, inhibitors. Thus resolved 3-D structure of the human DPP III-tynorphin complex enabled the design of the first transition-state peptidomimetics inhibitors, effective in the low micromolar range and resistant to proteolytic attack by DPP III. In addition, the combination of experimental and in silico study facilitated the discovery of new molecules with improved inhibitory activity, as was indicated by the QSAR analysis of coumarin derivatives.

A new direction in the field is the development of fluorescent inhibitor for monitoring the DPP III activity. The results obtained with the pyrene- or cyanine-guanidiniocarbonyl-pyrrole conjugates are promising in this respect, as these compounds show strong fluorescence upon binding to DPP III and potent inhibition of this enzyme (K_i_ value 0.2–0.3 µM).

Most recently overexpressed DPP III emerges as a potential drug target in several human pathologies, like cancer progression, sepsis and septic shock. To date, no inhibitors of DPP III have been evaluated in clinical trials. However, DPP III-blocking therapy has been shown to improve outcomes in preclinical sepsis models where inhibition of circulating DPP III by a specific antibody was obtained. To reveal new, or to prove yet indicated (patho)physiological, functions of DPP III, selective and strong inhibitors are needed. A successful example is the use of the highly potent inhibitor JMV-390 (IC_50_: 1.4 nM) to reduce intracellular DPP III activity which helped identification of DPP III as the angiotensin (1–7) degrading enzyme in human renal epithelial cells.

## Figures and Tables

**Figure 1 molecules-27-03006-f001:**
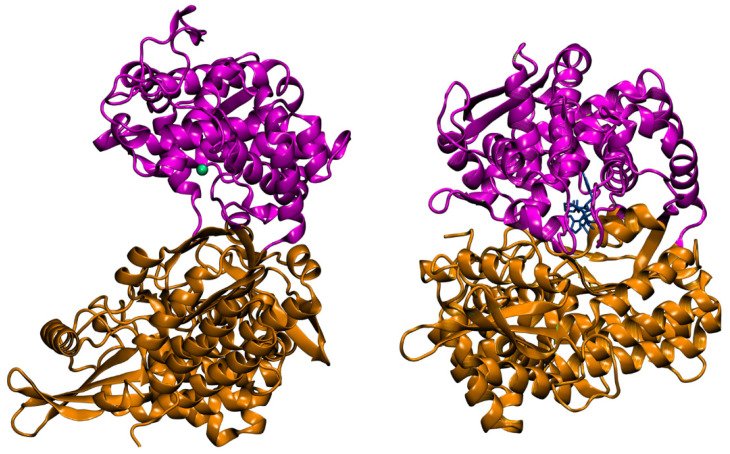
The 3-D structure of human DPP III without a ligand (PDB code: 3fvy) (**left**) and in its complex with pentapeptide tynorphin (PDB code: 3t6b) (**right**). The two domains of the DPP III protein are shown in magenta and brown, the zinc cation is represented as a green sphere, while the stick model shows the bound pentapeptide.

**Figure 2 molecules-27-03006-f002:**
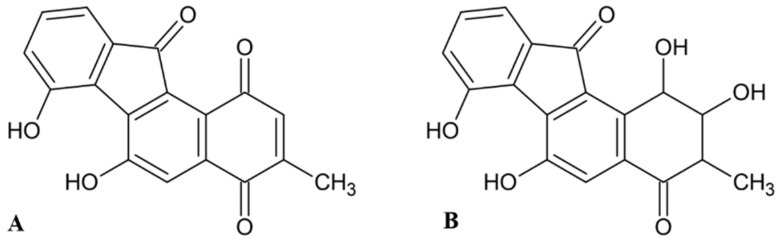
Structures of fluostatins (**A**) and (**B**).

**Figure 3 molecules-27-03006-f003:**
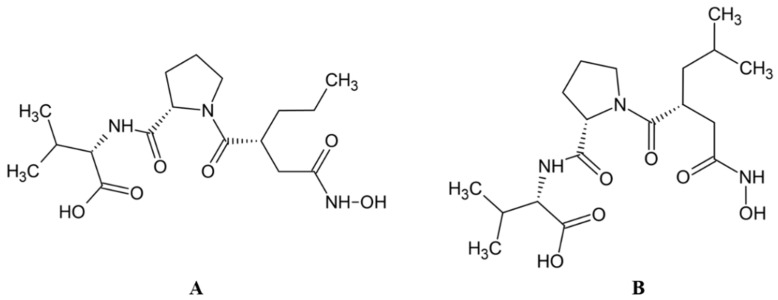
Structures of propioxatins (**A**) and (**B**).

**Figure 4 molecules-27-03006-f004:**
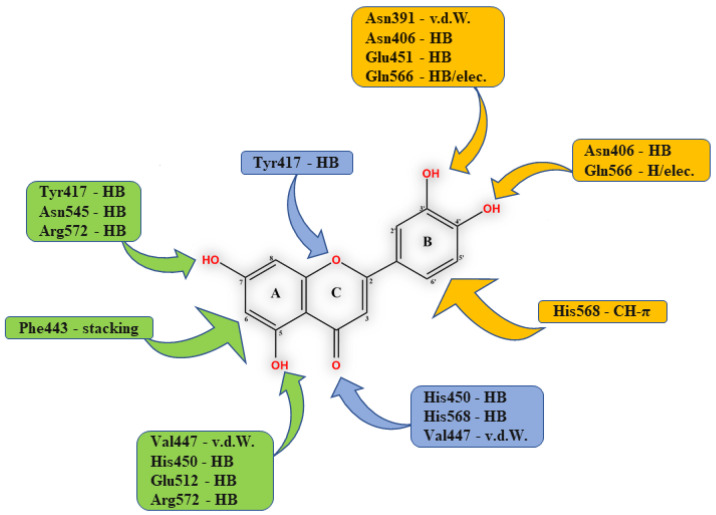
Schematic representation of the interactions between luteolin and amino acid residues in human DPP III binding site during 123 ns of MD simulations.

**Figure 5 molecules-27-03006-f005:**
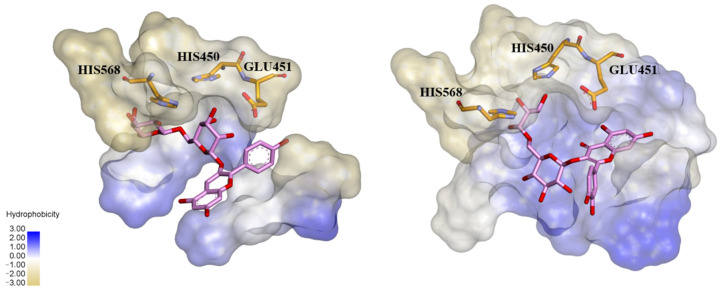
Binding mode of quercetin 3-O-rutinoside (**left**) and pelargonidin 3-O-robinobioside (**right**) into the binding pocket of human DPP III, as predicted by docking studies. Only residues essential for enzyme activity are presented. Hydrophobic surfaces are given according to the Ligplot^+^ diagrams [46,47].

**Figure 6 molecules-27-03006-f006:**
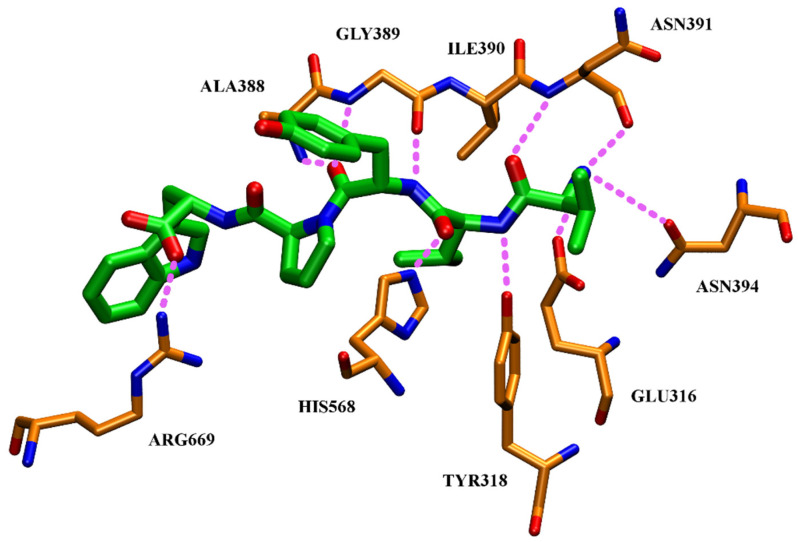
Polar interactions of tynorphin (green) with human DPP III [13].

**Figure 7 molecules-27-03006-f007:**
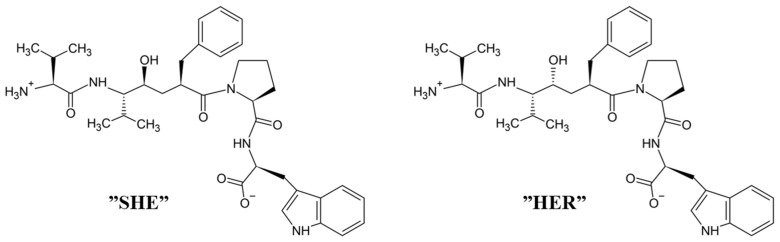
The (*S*)-hydroxyethylene transition state mimetic “SHE” and (*R*)-hydroxyethylene inhibitor “HER”.

**Figure 8 molecules-27-03006-f008:**
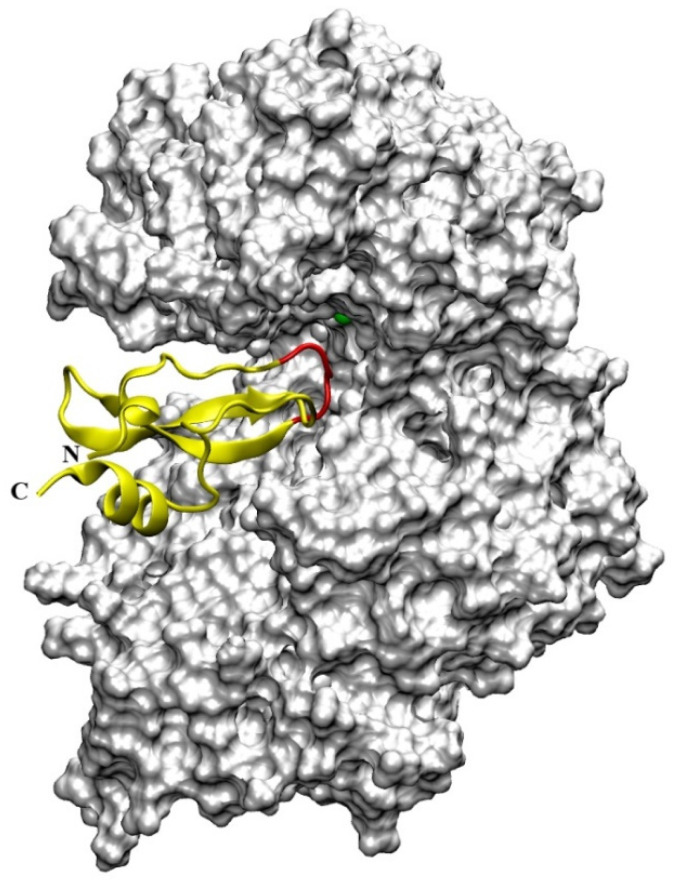
Overall structure of the human DPP III in complex with aprotinin (yellow) obtained by molecular docking. Aprotinin’s canonical binding loop is colored red, and zinc cation is represented as a green sphere.

**Figure 9 molecules-27-03006-f009:**
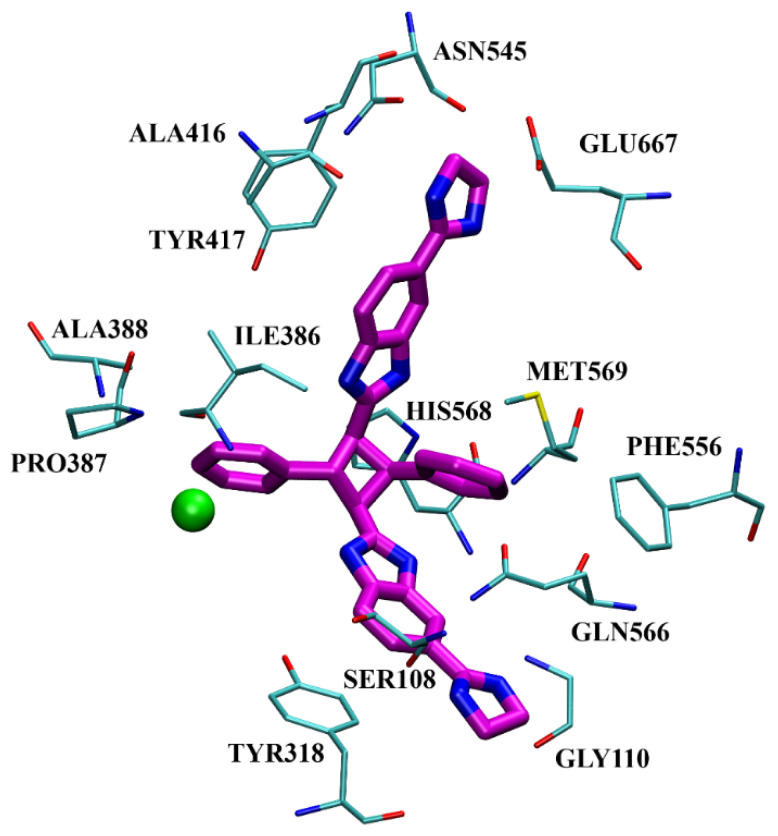
Interactions of inhibitor **1′** (magenta) in complex with human DPP III obtained after 40.8 ns of MD simulations. The Zn^2+^ is presented as a green sphere.

**Figure 10 molecules-27-03006-f010:**
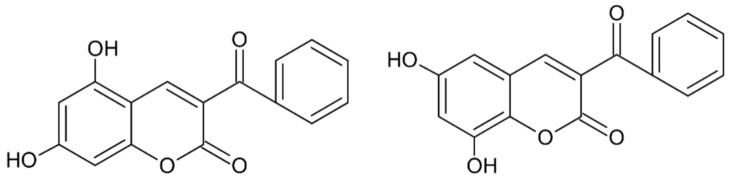
Structures of proposed coumarin derivatives with improved inhibitory activity towards human DPP III.

**Figure 11 molecules-27-03006-f011:**
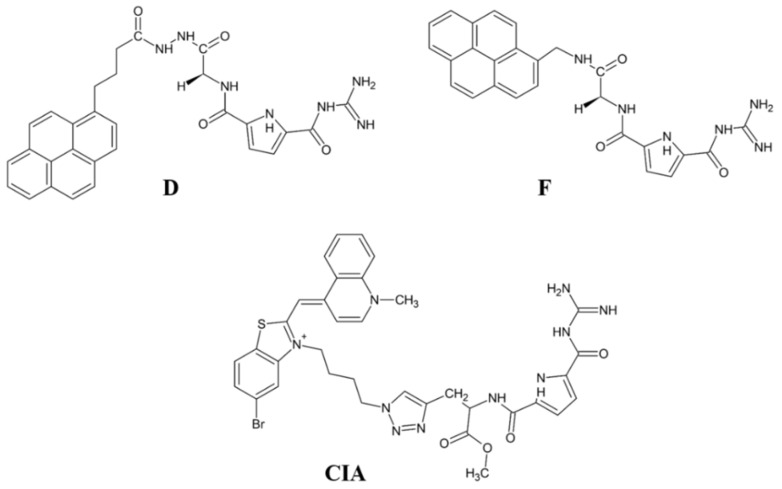
Structure of GCP conjugates **D**, **F** and CIA.

**Figure 12 molecules-27-03006-f012:**
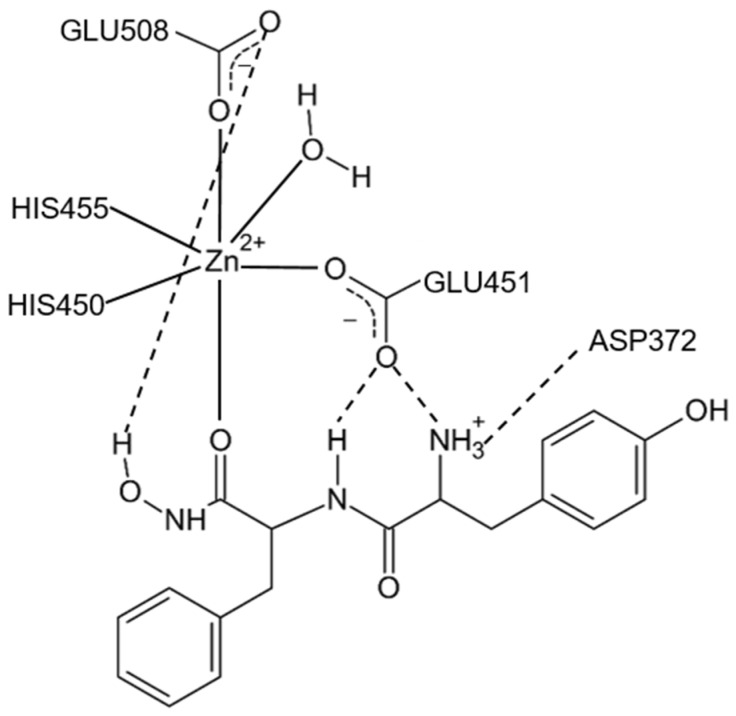
Schematic representation of the human DPP III active site with bound inhibitor Tyr-Phe-NHOH. Adapted from [14].

**Figure 13 molecules-27-03006-f013:**
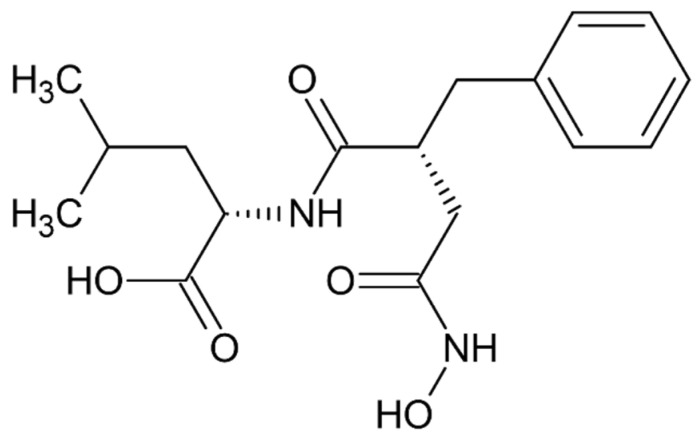
Structure of JMV-390.

**Table 1 molecules-27-03006-t001:** Chemical structures of the flavonoids and their effects on human DPP III activity. Adapted from [45].

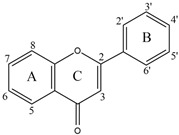		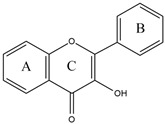		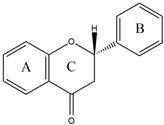			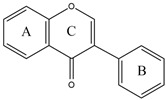	
Flavone		Flavonol		Flavanone			Isoflavone	
Group	Name	Substitution						IC50 (μM)
		2′	3′	4′	5	6	7	
Flavone	Luteolin	H	OH	OH	OH	H	OH	22.0
	Apigenin	H	H	OH	OH	H	OH	54.0
	6-Hydroxyflavone	H	H	H	H	OH	H	82.2
	Chrysin	H	H	H	OH	H	OH	123.1
	Flavone	H	H	H	H	H	H	292.5
Flavonol	Galangin	H	H	H	OH	H	OH	23.4
	Fisetin	H	OH	OH	H	H	OH	24.9
	Kaempferol	H	H	OH	OH	H	OH	32.9
	3,6-Dihydroxyflavone	H	H	H	H	OH	H	56.4
	Quercetin	H	OH	OH	OH	H	OH	74.1
	Morin	OH	H	OH	OH	H	OH	85.0
	3,7-Dihydroxyflavone	H	H	H	H	H	OH	98.0
	3-Hydroxyflavone	H	H	H	H	H	H	188.2
Flavanone	Flavanone	H	H	H	H	H	H	437.2
Isoflavone	Genistein	H	H	OH	OH	H	OH	36.6

**Table 2 molecules-27-03006-t002:** The selected intermolecular hydrogen bonds (HB) and hydrophobic interactions (HI) averaged over 200 ns long MD simulations of the human DPP III—aprotinin complex [55]. Corresponding human DPP III substrate binding subsites are given in brackets.

Aprotinin Residue		Human DPP III Residue		HB (%)	HI (%)
Asp3	side	Ser497	side	7.5	-
Glu7	side	Ser500	side	-	28.5
Pro8	side	Thr501	side	-	12.2
Pro8	main	Ser504 (S2)	side	1.2	-
Tyr10	side	Ser504 (S2)	side	-	74.9
Tyr10	side	Ala567	side	-	60.9
Gly12	main	His568 (S1′, S2′)	side	2.6	6.7
Pro13	side	Leu413	side	-	4.4
Pro13	side	Glu508 (S2, S1)	side	-	1.0
Cys14	side	Ala388 (S1′,S2′, S3′)	side	-	35.1
Lys15	main	Ala388 (S1′,S2′, S3′)	main	88.3	-
Lys15	main	Ala416 (S3′)	main	5.2	-
Ala16	side	Gly385	main	-	32.1
Arg17	side	Ser101	main	59.3	-
Arg17	main	Ser108	side	5.7	-
Arg17	side	Phe109 (S2′)	side	-	41.5
Arg17	side	Ser384	main	74.6	-
Arg17	main	Gly385	main	75.2	-
Arg17	side	Ile386 (S3′)	side	-	90.7
Ile18	side	Glu316 (S2)	side	-	88.8
Ile18	side	Tyr318 (S1, S2′)	side	-	43.4
Ile18	side	Gly385	main	1.0	64.3
Ile18	side	Pro387 (S1, S1′, S2′)	side	-	21.5
Ile19	side	Phe109 (S2′)	side	-	18.6
Ile19	main	Tyr318 (S1, S2′)	side	2.5	-
Arg20	side	Glu316 (S2)	side	47.8	-

**Table 3 molecules-27-03006-t003:** Inhibitory effect of various benzimidazole derivatives on the activity of human DPP III.

Compound No.	Structure	IC_50_ (µM)
1	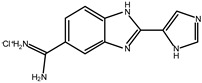	53
2	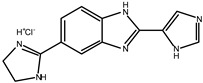	18
3	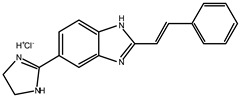	~10
4	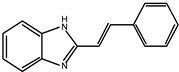	>100

**Table 4 molecules-27-03006-t004:** Inhibitory effect of 2,4-disubstituted-1,3-di-(5-amidino-2-benzoimidazolyl)-cyclobutane hydrochlorides on the activity of human DPP III. Adapted from [57].

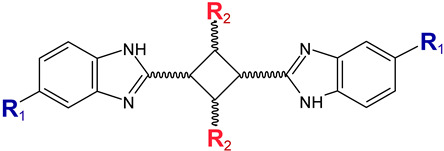
Compound No.	R_1_	R_2_	IC_50_ (µM)
**1′**	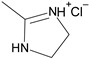	Phenyl	2.8
2′	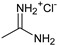	Phenyl	5.6
3′	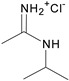	Phenyl	>10
4′	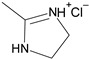	o-Cl-phenyl	1.7
5′	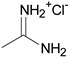	o-Cl-phenyl	~7
6′	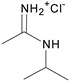	o-Cl-phenyl	~6
7′	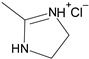	2-Furyl	~8
8′	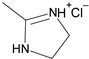	2-Thienyl	~10

**Table 5 molecules-27-03006-t005:** Structures of coumarin derivatives and values of experimentally determined inhibition of human DPP III (at 10 µM concentration of compounds) [60]. Numbers in brackets represent IC_50_ values; NA, no activity.

	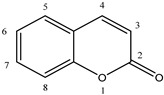	
Compound No.	Substituents	DPP III inh. (%)
1	3-acetyl; 6-bromo	28.5
2	3-acetyl; 6-hydroxy	12.8
3	3-acetyl; 7-diethylamino	NA
4	3-acetyl; 7-hydroxy	16.2
5	3-acetyl; 8-ethoxy	NA
6	3-acetyl; 8-hydroxy	NA
7	3-acetyl	7.8
8	3-benzoyl; 6-chloro	4.4
9	3-benzoyl; 6,8-dibromo	NA
10	3-benzoyl; 6-hydroxy	67.5
11	3-benzoyl; 7-benzoyl	22.8
12	3-benzoyl; 7-hydroxy	100 (1.10 µM)
13	3-benzoyl; 7-methoxy	16.5
14	3-benzoyl; 8-ethoxy	NA
15	3-benzoyl	9.6
16	3-cyano; 6-bromo	7.9
17	3-cyano; 6-methoxy	19.8
18	3-cyano; 6-hydroxy	44.6
19	3-cyano; 7-benzoyl	7.1
20	3-cyano; 7-methoxy	NA
21	3- cyano; 8-hydroxy	62.6
22	3-cyano; 8-ethoxy	NA
23	3-cyano	NA
24	3-ethoxycarbonyl; 6-bromo	NA
25	3-ethoxycarbonyl; 6-chloro	20.1
26	3-ethoxycarbonyl; 6-dihydroxyamino	59.7
27	3-ethoxycarbonyl; 6-hydroxy	66.0
28	3- ethoxycarbonyl; 6,8-dibromo	29.4
29	3-ethoxycarbonyl; 7-methoxy	NA
30	3-ethoxycarbonyl; 8-ethoxy	NA
31	3-ethoxycarbonyl	NA
32	3-methoxycarbonyl; 6-bromo	6.5
33	3-methoxycarbonyl; 6-dihydroxyamino	21.2
34	3-methoxycarbonyl; 6-hydroxy	23.5
35	3-methoxycarbonyl; 6-methoxy	9.9
36	3-methoxycarbonyl; 7-hydroxy	100 (2.14 µM)
37	3-methoxycarbonyl; 7-methoxy	NA
38	3-methoxycarbonyl	2.3
39	coumarin	NA
40	7-hydroxycoumarin	2.1

**Table 6 molecules-27-03006-t006:** Inhibition of human DPP III by dipeptidyl hydroxamic acids at pH 7.4 and 8.0.

Inhibitor	K_i_ (µM) at pH 7.4	K_i_ (µM) at pH 8.0
H-Phe-Phe-NHOH	0.028	0.11
H-Phe-Leu-NHOH	0.65	1.24
H-Phe-Gly-NHOH	4.63	14.51
H-Tyr-Phe-NHOH	0.030	0.103

## Data Availability

Not applicable.

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
