# Peer review of "Survey of Dipeptidyl Peptidase III Inhibitors: From Small Molecules of Microbial or Synthetic Origin to Aprotininâ€"

_molecules, 2022, doi:10.3390/molecules27093006_

Round 1
Reviewer 1 Report
In the submitted review manuscript for evaluation (Manuscript ID: molecules-1684985), the authors elaborated on the literature data on natural and synthetic inhibitors of dipeptidyl peptidase III, a biochemically very important cytosolic metallopeptidase.
All praise to the authors. The text is excellently and logically conceived, nicely written, with particular reference to the contribution (as I understood it) of the research group they are now members of.
Although Einstein once said that "Aesthetics should be left to tailors", except for one suggestion for the introductory part of the paper (1), my remarks (2-7) are pretty minor, and they are all related to technical aspects ("Aesthetics").
- I think it would be helpful for the reader to give a more detailed description of the structure of both DPP III domains (i.e., elements of secondary and ): lines 59-61.
- Please put ALL words of Latin origin in italic (lines): 38, 116, 415, 573. Further, put 2+ in superscript (line 49), 50 in subscript (line 82), replace (last column number in Table 2) 47,8 with 47.8.
- I suggest that the authors uniform all over the text quoting values for different constants, in the sense of putting a sign between the symbol for the constant and the corresponding value, as in other parts of the text: so, add either ":" or "=" or "of" or "value"! These are the values on (lines): 140, 158, 276, 444, 445, 446, 458, 565, 569, 593.
- One space of surplus or deficit can be found in several places in the text (lines): 80, 83, 110, 162, 209, 212, 242, 275, 344, 506, 523, 546.
- Does "it" (line 463, new paragraph!) refer to compound 12? Please clarify.
- Unify using the tab key in abbreviations and ACRONYMS list!
- The zinc cation is represented in one figure by a blue sphere (Fig. 1), and in the other two (Figs. 8 and 9) by a green one. Eventually, unify; I don't insist on that.
Author Response
REVIEWER 1
Comments and Suggestions for Authors
In the submitted review manuscript for evaluation (Manuscript ID: molecules-1684985), the authors elaborated on the literature data on natural and synthetic inhibitors of dipeptidyl peptidase III, a biochemically very important cytosolic metallopeptidase.
All praise to the authors. The text is excellently and logically conceived, nicely written, with particular reference to the contribution (as I understood it) of the research group they are now members of.
Although Einstein once said that "Aesthetics should be left to tailors", except for one suggestion for the introductory part of the paper (1), my remarks (2-7) are pretty minor, and they are all related to technical aspects ("Aesthetics").
- I think it would be helpful for the reader to give a more detailed description of the structure of both DPP III domains (i.e., elements of secondary and ): lines 59-61.
- Please put ALL words of Latin origin in italic (lines): 38, 116, 415, 573. Further, put 2+ in superscript (line 49), 50 in subscript (line 82), replace (last column number in Table 2) 47,8 with 47.8.
- I suggest that the authors uniform all over the text quoting values for different constants, in the sense of putting a sign between the symbol for the constant and the corresponding value, as in other parts of the text: so, add either ":" or "=" or "of" or "value"! These are the values on (lines): 140, 158, 276, 444, 445, 446, 458, 565, 569, 593.
- One space of surplus or deficit can be found in several places in the text (lines): 80, 83, 110, 162, 209, 212, 242, 275, 344, 506, 523, 546.
- Does "it" (line 463, new paragraph!) refer to compound 12? Please clarify.
- Unify using the tab key in abbreviations and ACRONYMS list!
- The zinc cation is represented in one figure by a blue sphere (Fig. 1), and in the other two (Figs. 8 and 9) by a green one. Eventually, unify; I don't insist on that.
Authors Reply to Reviewer's Comments:
We are thankful to Reviewer 1 for critical and careful reading of our manuscript. We have accepted all comments and suggestions.
Reply to Comment 1:
We agree that more details on the structure of DPP III should be given. Therefore, following text was added (lines 62-72) in revised manuscript: “The zinc-binding site and the catalytically important residues are located in the upper lobe, which is mostly α-helical. The lower lobe besides α-helices contains a smaller β-sheet portion (a five-stranded β-barrel). The two domains are connected by a number of loop regions. The first 3-D structure of human DPP III in complex with pentapeptide revealed a huge domain motion resulting in the complete closure of the cleft around the bound peptide substrate [13] (Figure 1). The active-site zinc ion is coordinated by the two histidine residues that belong to the conserved HEXXGH motif, a second glutamic acid residue from the conserved sequence motif EEXR(K)AE(D) and by the water molecule. In the structure of human DPP III, zinc-ligands are His450, His455 and Glu508. The Glu from the first zinc-binding motif (Glu451 in human DPP III) is proposed to act as a general base activating the water molecule which attacks the scissile peptide bond [12,16].”
Reply to comments 2 to 4: All suggested corrections are included in the revised manuscript.
Reply to comment 5: "it" (line 463 in submitted manuscript) refers to compound 12. In revised form, “it” is replaced by “compound 12” (line 474).
Reply to comment 6: Abbreviations and Acronyms list is corrected.
Reply to comment 7: We agree that this colour should be unified. Therefore, a new Figure 1 has been prepared for the revised manuscript, with the zinc cation represented by a green sphere.

Reviewer 2 Report
The review focuses on the dipeptidyl peptidase III inhibitors. The article is very well written and organized. The proper subgrouping of the information eases the understanding. I only have few minor comments. The manuscript can be accepted for publication if these are taken care of.
- For the sake of readers, I would suggest defining the terms which are relevant in inhibitory effects such as IC50, Ki. As the paper focuses on these terms, a definition would be helpful.
- Does reference [19] talk about all the inhibitors mentioned in that section? If not, please include other references.
- The structure figures look nice. I am curious, is there any reason you selected few of them to include and not others? On which basis did you choose?
- Section 4.(.poly)…. Change to section 4.(poly)
Author Response
REVIEWER 2
Comments and Suggestions for Authors
The review focuses on the dipeptidyl peptidase III inhibitors. The article is very well written and organized. The proper subgrouping of the information eases the understanding. I only have few minor comments. The manuscript can be accepted for publication if these are taken care of.
- For the sake of readers, I would suggest defining the terms which are relevant in inhibitory effects such as IC50, Ki. As the paper focuses on these terms, a definition would be helpful.
- Does reference [19] talk about all the inhibitors mentioned in that section? If not, please include other references.
- The structure figures look nice. I am curious, is there any reason you selected few of them to include and not others? On which basis did you choose?
- Section 4.(.poly)…. Change to section 4.(poly)
Authors Reply to Reviewer's Comments:
Reply to comment 1: We agree that these terms should be defined. In the revised manuscript one sentence with definitions of IC50 and Ki is added at the end of Introduction (lines 137-139): “The inhibitor potency is expressed as the IC50 value, defined as the concentration of an inhibitor which caused 50% reduction of the enzyme activity, or by the inhibition constant Ki, the equilibrium dissociation constant of the enzyme-inhibitor complex.”
Reply to comment 2: Besides the reference [19], references [2], [17], [18], [20], [21] and [22] are mentioned related to the inhibitors in that section (lines 78-99 in revised manuscript).
Reply to comment 3: We tried to encompass all groups of DPP III inhibitors in our review and to represent the structures of the most potent inhibitors, or at least the general structures for the group (e.g. flavonoids, coumarins). Regarding the DPP III structure, we have chosen two crystal structures of human DPP III that were resolved first: ligand-free (PDB code: 3fvy) and in complex with tynorphin (PDB code: 3t6b), which also represents the binding mode of peptide substrates to this enzyme.
Reply to comment4: This is corrected.